# Family Caregivers’ Experiences during the COVID-19 Pandemic: Qualitative Study

**DOI:** 10.3390/healthcare12100970

**Published:** 2024-05-08

**Authors:** Milagros Rico-Blázquez, Raquel Sánchez-Ruano, Cristina Oter-Quintana, Elena Polentinos-Castro, Ángel Martín-García, Pedro Otones-Reyes, Damián González-Beltrán, Mercedes Martínez-Marcos

**Affiliations:** 1Research Unit, Primary Care Assistance Management, Madrid Health Service, 28035 Madrid, Spain; elena.polentinos@salud.madrid.org; 2Research Network on Chronicity, Primary Care and Health Promotion—RICAPPS-(RICORS), ISCIII, 28035 Madrid, Spain; 3Gregorio Marañón Health Research Institute, Madrid Health Service, 28009 Madrid, Spain; 4Doctoral Program in Epidemiology and Public Health (Interuniversity), Rey Juan Carlos University, Alcorcón, 28922 Madrid, Spain; 5Nursing Department, Faculty of Nursing, Physiotherapy and Podiatry, Complutense University of Madrid, 28040 Madrid, Spain; angel.martin@salud.madrid.org; 6Ciudad de los Periodistas Healthcare Centre, Primary Care Assistance Management, Madrid Health Service, 28034 Madrid, Spain; rsruano@salud.madrid.org; 7Nursing Department, Faculty of Medicine, Autonomous University of Madrid, 28049 Madrid, Spain; cristina.oter@uam.es (C.O.-Q.); mercedes.martinezmarcos@uam.es (M.M.-M.); 8Nursing and Healthcare Research Group, IDIPHISA, 28222 Madrid, Spain; 9Preventive Medicine and Public Health Area, Health Sciences Faculty, Universidad Rey Juan Carlos, Alcorcón, 28922 Madrid, Spain; 10San Blas Healthcare Centre, Primary Care Assistance Management, Madrid Health Service, Parla, 28980 Madrid, Spain; 11San Andrés Healthcare Centre, Primary Care Assistance Management, Madrid Health Service, 28021 Madrid, Spain; pedro.otones@salud.madrid.org; 12Barrio del Pilar Healthcare Centre, Primary Care Assistance Management, Madrid Health Service, 28029 Madrid, Spain; damianglezbel@gmail.com

**Keywords:** family caregivers, primary healthcare, nursing, person-centered care, COVID-19, pandemics, life change events, coping strategies, qualitative research

## Abstract

**Background:** The COVID-19 pandemic imposed lockdown measures that affected caregiving. Understanding caregivers’ context provides reveals their adaptive strategies to continue caring in this situation of uncertainty and isolation. **Objective:** To better understand the caregiving experiences of caregivers looking after dependent individuals living in the community during the pandemic. **Design:** Qualitative research, phenomenological approach. **Setting:** Primary healthcare centers in Madrid region (Spain). **Participants:** 21 family caregivers. **Methods:** Purposive and theoretical sampling was used to recruit caregivers across nurses from primary healthcare centers. Participants were interviewed using a semi-structured interview guide to explore the caring experience. Interview transcripts were evaluated using thematic analysis. **Results:** The findings were categorized into two themes: “Caregivers during lockdown—providing care in a time of adversity” and “Caregiving toward normality”. The sub-themes identified were the re-structuring of before-care services and the introduction of new care approaches, managing the dependent person’s health problems, looking after oneself, and dealing with adversity. To adapt to the new normal, strategies were put in place designed to recover confidence and trust, reincorporate assistance, and reconnect with others. **Conclusions:** Care intensified during the pandemic. Caregivers took on the task without assistance, focusing on preventing contagion and protecting themselves to be able to continue giving care.

## 1. Introduction

Alongside other countries in the Mediterranean region, Spain has one of the longest life expectancies; currently 3.4% of the population is over 75 years of age [1]. While local, regional, and national governments have programs dedicated to caregiving for dependent people [2], a large part of care falls within the family. It is estimated that caregivers dedicate over 20 h a week to this activity [1]. 

The care of elderly dependent people within the family follows an inter-generational paradigm in which the necessary burden of attention falls on younger people, and an intra-generational paradigm in which care is provided by people of a similar age [3,4]. The profile of a caregiver in Spain is that of a woman, normally the partner or daughter, with a basic level of education who dedicates most of her time to domestic labor [5,6,7]. In recent years, we have seen a trend toward a greater balance in terms of gender equality [1]. 

The effects on family caregivers of providing care has been widely studied, with it described as having a negative impact on their health and leading to a poorer quality of life, fewer opportunities for social relationships, and employment problems [8,9,10]. To tackle this impact, caregivers develop strategies that include altering the way that they approach relationships, interact with other family members, adapt their homes, and look for spaces in which they can disconnect from their role as caregiver [11]. 

The COVID-19 pandemic has seen a revolution in almost every aspect of people’s lives. The importance of family caregivers and how women continue to shoulder most of the responsibilities in this area has become abundantly clear [12]. 

On 15 March 2020, the Spanish government declared a full lockdown, with only those deemed to be working in essential jobs allowed to leave their homes [13]. From 21 June 2020 onward, there was a partial relaxing of the rules as the epidemiological situation began to improve, marking the start of the so-called “new normal” [14]. On 27 December 2020, the vaccination campaign was launched, ensuring that all aged 65 or over had received at least one dose by April 2021 and 50% of the whole population by 22 June 2021 [15]. 

During lockdown and for a significant part of this new normal, there was a re-structuring of the health and social systems that caused problems in terms of access to healthcare and the breaking down of support networks, with a parallel increase in the intensity of caregiving and the responsibilities it entailed in many homes [16]. The relatives of those who were most vulnerable had to spend all their time at home, as the only people who were responsible for caring duties as well as looking after others who were not always fully prepared for this attention [17]. 

Research into the effects of the pandemic on the family caregivers’ health shows that they are more likely to experience stress, overwork, social isolation, and mental health problems [18]. In the case of long-term caregivers and those looking after patients with dementia and other neurodegenerative diseases, somatic problems are more common [19,20]. Other research also highlighted the impact on emotions related to anguish, impotence, insecurity, fear, worry, and sadness [21]. In order to confront these stressors and attempt to reduce the impact of the pandemic on their health and their role as providers of care, caregivers implemented strategies that included adapting to the available support and the new ways of communicating with health service professionals [22], although no evidence has been found that these strategies have been explored in depth. 

The COVID-19 pandemic developed over the course of an extensive time axis, during which the ever-changing epidemiological situation was marked by frequent alterations to the restrictions of movement, access to health and social services, and changes in measures introduced to prevent further contagion. 

Through our research, we have sought to better understand the caregiving and self-care experiences of family caregivers looking after their relatives that live in the community during the COVID-19 pandemic. 

A holistic and contextual understanding of the informal caregiving experience in a high-stress and isolating context can strengthen the nurses’ ability to engage caregivers in the design of nursing care.

## 2. Materials and Methods

### 2.1. Research Team and Reflexivity 

This study was undertaken by a team of seven nurses and one physician (five women and three men) working in the clinical practice, teaching, and research fields. The research work of two of the members of the team mainly focused on family caregivers looking after dependent people. The rest of the team has prior experience of providing healthcare for dependent people and caregivers within the primary healthcare system. During the research process, the members of the team paid special attention to the way in which their prior clinical practice, research, and personal experience might affect the conceptualization, design, and gathering and analysis of data. These questions were addressed at team meetings, paying special attention to the preconceived notions that stem from the shared experiences that determine analytical rigor. All the researchers took part in the various stages of the process. The interviewers had no prior relationship with participants, introducing themselves by name and outlining their academic background at the start of the interview. 

### 2.2. Objective 

To better understand the caregiving experiences of family caregivers looking after the dependent people that lived in the community during the COVID-19 pandemic. 

### 2.3. Design 

We conducted a qualitative research design with a phenomenological approach, enabling the study of the phenomenon within its natural context to derive interpretations based on the participants’ meanings [23,24,25]. The COREQ checklist is available as Appendix information (Appendix A). 

### 2.4. Setting 

The study was held in the Madrid region. Participants were recruited at primary healthcare centers. 

### 2.5. Participants and Sampling

The participants in the research were family members who care for dependent relatives. The inclusion criteria required that participants be family caregivers or a next-of-kin who lived with the dependent person for at least six months before the start of the pandemic, be 18 or older, and voluntarily agree to take part in the study. Exclusion criteria ruled out caregivers who had been in a grieving process that had lasted at least a year, those diagnosed as having a serious mental health condition, or those who were looking after patients undergoing active oncological treatments. 

The selection of study participants was sequential, corresponding to the different waves of data collection. Initially, sampling was intentional to maximize diversity. Caregivers with extensive and rich experiences were selected, considering factors such as their experiential knowledge of the study topic and their willingness and ability to discuss it.

As the analysis progressed, participants’ profiles were added from a diverse range of situations, including variations in age, length of caregiving, illnesses of the dependent person and caregivers, level of dependency of the cared-for person, caregiver’s occupational activity, and the kind of assistance provided to the patient during the pandemic. 

The selection and recruitment of participants took place both in nursing consultations and in the homes of the dependent persons. Specifically, the recruitment was conducted by the reference nursing professional assigned to each caregiver. A total of 36 individuals were informed of the purpose of the research, invited to participate, and asked for their permission to provide their telephone number to the research team interviewers. Five caregivers declined to participate, and of the thirty-one who consented, five could not be contacted. Unfortunately, one of them did not carry out the interview due to the death of the dependent person. A total of 25 interviews were carried out, of which 21 were included in the subsequent analysis. Figure 1 shows a flow diagram of participants in the research. 

The majority of the caregivers interviewed were women, the daughters of the dependent person. The ages of the participants ranged from 28 to 79, with an average age of 60. Very few of them were unemployed. They had spent an average of about six years providing care. Table 1 describes the characteristics of the interview participants. 

### 2.6. Data Collection 

Data were gathered through semi-structured interviews held between June 2021 and January 2022 in three phases. Based on existing literature, the research team drew up an interview guide (Appendix B) with broad open-ended questions which sought to explore their experience as caregivers during the period of strict lockdown and the so-called ‘new normal’. The initial script was amended in each phase as the analytical process moved forward, incorporating new questions in order to fine-tune the emerging themes. Given the situation caused by the pandemic, participants were offered the option to carry out the interview by phone on the date and at the time of their choosing. Ten participants opted for a phone interview, while fifteen chose to conduct them face-to-face, either in their homes or in a public space such as parks and cafes. During the interviews, only the interviewer and interviewee were present, except in those cases in which the caregiver was unable to delegate care of the dependent person, and therefore, they also had to be present. The interviews, which were digitally recorded, lasted between 45 and 70 min, with field notes taken during and after. A transcript was made of the recording, which were collated with the others. As they were deemed to be faithful to the source material, there was no need to corroborate this with the interviewees or to re-interview the participants, although in some cases, standard punctuation norms were applied to improve legibility. The written and verbal consent of each participant was sought for the interview and the audio recording. The gathering of data continued until theoretical saturation was reached [26]. 

### 2.7. Data Analysis 

Thematic analysis was undertaken following the phases suggested by Braun and Clarke [24]. In the first phase of familiarization, the researchers listened to the recordings, read the transcript separately, and made the necessary corrections, taking notes on their first impressions resulting from the data. In the second phase, the eight members of the team worked in pairs to identify the preliminary codes such as “assuming new care” and “shutting oneself away”, taking notes on the emerging themes. Appendix C offers a summary of this encoding. An inductive code tree was prepared to organize the information that was developed in group discussion. In phase three, relevant themes were jointly identified through the relationships between preliminary code groups, describing topics such as “intensification of family care”. In phase four, themes were closed, reviewed, and refined in order to avoid overlapping. Themes were defined during phase five, with transcriptions reviewed to ensure coherence with the general argument of the analysis undertaken and with each topic as described. In the last phase, the final report was drawn up and discussed. During the process, the researchers prepared a methodological manual that brought together the critical reflections and analytical inferences that had arisen, which were discussed at group sessions. 

### 2.8. Ethical Approval 

All participants provided written informed consent. The research was approved by Madrid Primary Healthcare Network Management’s Central Research Commission (14/21) and by the Clinical Research Ethics Committee of Hospital Universitario Puerta de Hierro (07/690331.9/21). The research respected the basic ethical principles of autonomy, justice, beneficence, and non-maleficence, pursuant to the standards of Good Clinical Practice, the Declaration of Helsinki (Fortaleza 2013), and the 1997 Oviedo Convention. The processing, communication, or assignment of data was undertaken pursuant to currently applicable law (the Spanish Protection of Personal Data and Guarantee of Digital Rights Act, Organic Law 3/2018 of 5 December 2018) and with respect for the ARCO rights of access, rectification, cancellation, and objection, as provided for in the aforesaid act. 

### 2.9. Rigor 

The research followed the criteria of credibility, transferability, reliability, and confirmability proposed by Guba and Lincol [27]. The sampling strategy sought to incorporate people with a wide range of experiential characteristics that offered an in-depth perspective of the phenomenon being studied. A detailed description of the setting and the characteristics of the participants was drawn up. The research team sessions were recorded, and the theoretical–methodological decisions compiled into a research journal and made available to all team members. Every team member kept a personal journal in which the reflections that came up during the data gathering and analysis stages were recorded. The whole of the analytical process was undertaken in pairs. In order to validate the findings, all of the authors read the various research report drafts independently, discussing them before the final report was drawn up. 

The confirmability of the study was helped by the use of notes and discussions with colleagues and the research team on the various agreements and disagreements regarding the themes being analyzed until consensus was reached.

## 3. Results

Two central themes were identified: “Caregivers during lockdown—providing care in a time of adversity” and “Caregiving toward normality”, which describe the experiences of self-care and the family caregiver taking care of their relatives during the COVID-19 pandemic. Table 2 shows the identified themes and sub-themes. 

### 3.1. Caregivers during Lockdown—Providing Care in a Time of Adversity 

The limitations on mobility and restricted or suppressed access to social health services represented a reorganization and adjustment of family care activities, altering the previous care routine. Family caregivers compared this life with the life they had previously. This comparison created feelings of nostalgia for their previous lives, highlighting the fact that caregiving during the pandemic took place under much more complex conditions, putting their ability to adapt to the test. 


*Re-structuring family caregiving*


Caregivers who had been given part-time help did not receive it during the pandemic. Dependency support services were temporarily suspended by authorities who administered social healthcare management. Occasionally, the caregivers turned down this assistance, thus becoming the sole full-time caregivers. 


*“The biggest change was not being able to go out because my husband, who depends on me, in normal times goes to a day centre during the day. So, then he was at home from when he woke up to when he went to bed.”*
(CJ_RSR_1)


*“We took the precaution to turn down the little help we used to get because we didn’t want anybody passing anything on to us.”*
(SB_AMG_5)

The decision to decline the sharing of caregiving was based on a wish to preserve the lives of the carers and their families, foregoing anything that might represent the risk of contagion by the virus. Both the caregiver and relative would remain together for days and weeks, ruling out the possibility of the former having any time or space to themselves.


*“Before, I could combine caregiving and my social life.”*
(SAN_POR_2)

When caregivers employed people in their homes as caregivers, these remained, as they too would be confined where they worked, a situation that would not represent any extra risk for the dependent person. 

Caregivers who continued their employment opted to request a furlough or else reduce their working hours in order that they could combine their jobs with caring for their families and the dependent person. Occasionally, caregivers chose to work from home, or else this was the arrangement imposed by their workplaces.


*“I asked to work from home so that I could stay here all day because I couldn’t leave her on her own.”*
(CJ_RSR_10)

For other caregivers, despite not receiving any assistance, giving up their usual work schedule as a result of the lockdown came as a relief, as it meant that they could dedicate themselves to fulltime caring.


*“I was relaxed, maybe I was a bit worried about the pandemic and all that… still, it really suited me in that sense, I was pretty calm, I didn’t have to follow a timetable, and there wasn’t all that pressure.”*
(SAN_POR_7)

However, for some workers, leaving their jobs and not being with their colleagues during such a critical time stirred up feelings of guilt. 


*“When I stayed here, that was fine, I didn’t get infected, I didn’t infect my mum. But I still felt a bit guilty, like I had abandoned them (her former colleagues at the supermarket—an essential service).”*
(BP_DGB_4)

Key workers who carried out their jobs outside home had to think up ways of caring for their dependent relative in their absence. They had to be circumspect in the care they were giving, preparing the domestic environment and adapting it to the functional capacity of the person they were looking after in a way that they were able to satisfy their needs independently. Some caregivers even had cameras installed so they could monitor the situation while they were away.


*“I gave him lots of glasses of water, made sure the TV remote was by his side.”*
(CJ_RSR_6)


*“I installed a video surveillance camera; he also had the telecare call button… I could connect to an app from work; I’ve got a few cameras here at home so I could see if he got up, if he didn’t get up. I’m relying on that above all.”*
(SAN_POR-2)


*Assuming and incorporating new care*


The lack of assistance increased the workload of caregivers, who had to assume the full burden and incorporate new care. They had to undertake activities aimed at satisfying basic day-to-day life. They programmed entertainment manuals (drawing, writing, coloring, etc.). In the case of people suffering cognitive decline, these activities seemed to be orientated toward minimizing the possible effects of the lockdown on the advance of the illness and preventing episodes of agitation. On other occasions, they took part in activities organized by the same assistance services they had worked with previously, which often took place online. 


*“Well, you know, its personal hygiene—I wash her in the morning, comb her hair and so on. Then I have to help her shower and… That sort of thing. If she needs her toenails cutting, I do that. They’re the sort of things I do.”*
(CJ_RSR_5)


*“With my mother, I get her drawing, give her things to do.”*
(CJ_RSR_11)


*“The day centre gave me a lot of support; they sent me activities via WhatsApp.”*
(SAN_POR_7)

Caregivers also had to learn and incorporate new forms of care aimed at preventing contagion. Actions focused on extreme domestic cleaning of objects that had come in from outside, removing and cleaning garments that had been worn outside the home, and the frequent washing of one’s hands. 


*“Disinfecting the keys, the door, your clothes, taking off your shoes, washing your hands well… I disinfected everything.”*
(CJ_RSR_10)

Cleaning became an obsession, undertaken with extreme care and prudence.


*“I became hysterical about cleanliness, to the point where I had bleach diluted with water everywhere. I told the girls we had to clean absolutely everything that we touched, that the caregivers touched.”*
(CJ_RSR_8)

Family caregivers who had to keep working outside the home implemented actions related to hygiene and virus infection prevention in the exterior, altering their day-to-day habits. These actions included not sitting next to others or holding onto handrails on public transport. At home, preventative care in terms of infection was accompanied by physical separation and restricted contact with the dependent person, making distance a safeguard against transmission of the virus. 


*“I have to say, I was afraid too, taking the train, grabbing the handrails, because of course everyone was touching them.”*
(SB_AMG_8)


*Managing health problems*


Another of the central concerns of the caregivers was the supervision of their family’s healthcare processes. The collapse of the healthcare system obliged them to play a more active role in the supervision and monitoring of health, further adding to stress levels. Caregivers felt they had insufficient knowledge and were concerned by the potential damage that they might cause if they did not provide their care correctly. Managing the therapeutic routine, although guided by healthcare professionals, was therefore perceived as a further significant stressor. 


*“But yeah, it was a responsibility, which I see differently now, but at the time I felt really bad. Sometimes I couldn’t sleep, I kept thinking ‘What would happen if I got the dosage wrong, would they be like a vegetable or what?’ That’s the thing, there was nothing else I could do. The doctor told me to increase the dosage since I couldn’t talk to her every day.”*
(CJ_RSR_11)

Despite the reticence to have contact with others from outside the home, relatives and their caregivers were nonetheless open to access in the case of healthcare professionals. The same was not true for their friends and relatives. 


*“I wasn’t scared when the nurse came that he might infect us; I had to trust the health system: He told us that we had to put our facemasks on. Therefore, we did. We put on masks, gloves, used sanitiser gels… So, I was never afraid. I had complete trust in the nurse.”*
(SB_AMG_1)


*“The sacrifice came at Christmas and birthdays; you couldn’t be with your children. Everybody, every one of our children stayed in their homes and us in ours so that nobody was in danger.”*
(SB_AMG_1)

Sometimes courses of action were implemented due to COVID-19 infection, either in the caregiver or in a family member. To do so, they monitored any possible signs of alarm, such as rising temperatures, oxygen saturation, etc.


*“I monitored his vital signs; he didn’t have a fever…”*
(SB_AMG_8)


*Staying healthy and looking after oneself*


Staying healthy meant making an effort to look after oneself, something that is by no means easy in a situation of permanent strain, under the premise of not falling sick due to the fear of being unable to continue to give care.


*“Yeah, at night I was scared. I’m going to end up… And I ended up… My throat was hurting a lot… yeah, I caught COVID. I don’t know how but I got it. Maybe I’m going to die, because I didn’t know how this was going to turn out… should I have a fever? Am I going to feel really ill? I thought I’m going to fall asleep and then I’ll die. And my mother, what’s going to happen to her?”*
(CJ_RSR_11)

They took action designed to protect their role as a caregiver, seeking out the best possible way of safeguarding their needs. They therefore started doing physical exercise routines adapted to the context of the home and took up new free time and leisure activities such as cooking, reading, and listening to music. 


*“We used to walk about the house—we moved the table, the chairs, the sofa and anything that might get in the way and started walking around the house to keep in shape”*
(SB_AMG_1)

Some caregivers sought to compensate for the limitations placed on physical contact with family and friends through digital meetings that allowed them to keep in touch and comforted them. These calls allowed caregivers to actively engage in their roles as mothers, grandmothers, and friends, providing a sense of continuity and connection despite physical separation. By participating in these video calls, caregivers were able to fulfill their roles within the family unit, fostering a sense of belonging and preserving their social identity within their familial roles.


*“We would speak to each other [family members] on video calls.”*
(CJ_RSR_13)

In order to guarantee that their needs and those of their relatives were met, caregivers made use of informal community help networks that emerged spontaneously to provide food and medicine, shop delivery services, etc. These networks were an essential means of support that allowed people to continue providing more intensive care without them having to leave the home or leave the dependent person alone at any time. 


*“I was lucky that a family lives opposite who are very close to us, they’re also Peruvians. One of them helped me do the shopping. We’d communicate by phone, by WhatsApp. Then he’d come round and drop off the bag with the shopping.”*
(SAN_POR_7)


*Dealing with changes*


During the pandemic, caregivers found themselves up against an extraordinary situation that was characterized by great uncertainty and multiple stressors, such as not knowing how long things would continue, concern for relatives, and their employment and economic situation, all of which obliged them to adopt coping strategies. During the COVID-19 pandemic, caregivers were faced with a barrage of new and ever-changing information from the media. This bombardment of information added to the challenges they already faced, requiring them to adapt quickly to evolving guidelines and recommendations. 


*“Obviously, every day you would hear something, you wouldn’t know what to do. I was really terrified.”*
(SAN_POR_1)

This highlights the overwhelming and anxiety-inducing nature of the constant influx of information.

In order to confront the situation, some caregivers dedicated a large part of their time to reading and listening to the news, in order to better understand the situation, all of which helped them to take the steps they had to in order to not get sick. For other caregivers, the information overload in the media represented extra stress. 


*“There was a time during lockdown when we stopped watching television or listening to the radio. We only listened to music. I told my mother that I didn’t want to know anything about the news because it was always the same, the same, the same. All the figures they gave you were always the same. It tired you out in the end.”*
(CJ_RSR_8)

The normal measures that caregivers used to manage their stress and cope with the burden of work had to be interrupted. As one caregiver expressed, “I felt like there was no way out”. The caregivers who were able to temporarily give up their caring duties took the opportunity to disconnect.


*“I would make those moments (walking the dog) last longer so I could switch off from the girls, my mother and home life.”*
(CJ_RSR_13)

In order to be able to cope with the pandemic and look after the family, caregivers employed cognitive coping mechanisms. These strategies made them more resilient. Caregivers used cognitive strategies such as focusing on positive thoughts, maintaining an optimistic attitude, and practicing acceptance of the situation. These strategies helped them remain resilient amidst uncertainty and daily stress. 


*“I adapt my mind to the situation at hand as I know there is no alternative.”*
(SB_AMG_5)

Other caregivers found solace connecting with their spirituality. 


*“Prayer gives me great inner peace, a sense of calm.”*
(SAN_POR_7)

Without family support, some caregivers fantasized that someone would tell them that ‘everything was going to be OK’. However, the majority sought to overcome these emotional shortcomings by connecting to other people through their mobile phones and computers, allowing them to retain their ties and keep in touch with friends, relatives, and social networks.


*“I really wanted that person to come by and give me a hug (the neighbour who would bring the shopping round).”*
(SAN_POR_7)


*“I’d chat with my friends on WhatsApp, ‘Look at the cake I made! Look at this tart, I made that too!’ That’s what we used to do in the afternoons; I think it was a way of passing the time.”*
(CJ_RSR_1)

The physical and emotional strain experienced by the caregivers during lockdown had repercussions such as back pain, less time available for exercise, leisure, and rest, and impaired nutritional habits. 


*“The thing is, I’ve had backache for a month and a half now. I’m having problems with my sacral vertebrae; I’m going to start rehabilitation on the 31st. I can hardly lift my right arm because I’ve got a tendinitis that goes from my neck almost to me elbow.”*
(BP_DGB_4)


*“I’ve put on five kilos, for me it feels like I’ve been poisoned.”*
(SB_AMG_2)

Feelings of sadness, loneliness, anguish, and fear were channelled through a form of restrained venting as a way of finding some relief, without this impacting the person being looked after. 


*“Me, cry? I must admit I was going that way. But how could I cry in front of her? Of course I couldn’t.”*
(CJ_RSR_8)

### 3.2. Caring toward Normality


*Recovering confidence*


Once things began to open up after the strictest part of the lockdown, caregivers could gradually introduce changes, taking them towards the new normal. These changes were marked by the progress that was being made in terms of vaccinating the population and the reduced incidence of infection, which caregivers interpreted as a sign that allowed them to feel safer, despite the fear of contagion continuing and the danger that their family member might succumb to the illness. 


*“I don’t take my facemask off but I feel calmer, more relaxed.”*
(SB_AMG_8)


*“I think everything was a bit more relaxed after she had the first dose of the vaccine.”*
(BD_DGB_2)

Although social contact was opening up, caregivers continued to strictly apply their safety measures, such us observing mask wearing and preventative hygiene.


*“Yeah, that was what I thought; taking my mask off would have to wait a while. Others can say what they like; I’m going to carry on like before”.*
(SAN_POR_1)


*“The return to normal has been harder because in some ways you’re still afraid of becoming infected, right?”*
(CJ_RSR_9)


*The return of assistance*


Despite the fact that dependency was available again after the first weeks of lockdown, it was not until caregivers felt safer that they felt they could return to calling on home assistance and the reincorporation of dependent people to day centers. The return of available assistance freed caregivers of a lot of the burden of providing care on their own.


*“She’s been going to the day centre for a month and a half or so now, she couldn’t before. The truth is that it is a big relief for me because now I can do my activities; I can do the shopping and other simple things, right? If I go with her, I have to go a lot slower.”*
(CJ_RSR_11)


*Reconnecting with others*


Being allowed to go out again, along with the recovery of confidence and the reintroduction of assistance for those caring for family members helped caregivers to connect with the outside world and undertake activities away from the home, ensuring that they could gradually feel that they were getting their previous lives back with their old routines and escape routes. 


*“Being able to go out, go for a walk with your mask on, not getting too close to others but just a little… more like the old routine. I went to the beach in Alicante, where I usually go. I started going to the beach, with the special measures that didn’t allow certain things in the city, measures we had to comply with, but you know… at least I could finally get out of the neighbourhood, even if I had to wear a mask.”*
(CJ-RSR-1)

Getting back in touch with others took place gradually, with caregivers classifying their social contacts as being a lesser or greater risk of passing on infection. Social relations began with those who were deemed to be low risk: people who worked from home and who did not have teenage or young children. However, they continued to apply the recommendations on how to prevent contagion, keeping group relations small, meeting in open spaces, allowing home access to those relatives who presented the least risk, etc. 


*“The only one who came here to have lunch with us was the youngest, she worked from home. When she was on her own, she was on her own, so she came over to see us. If one of the children comes over, well they live in other houses, they go out to work.”*
(SB_AMG_2)


*“When we started up again, we went out to pavement bars for a drink but not at home.”*
(SB_AMG_2)

## 4. Discussion

### 4.1. Main Findings of the Study and Comparison with Other Studies 

The measures adopted during the COVID-19 health crisis had a significant impact on access to health and social services, their relationships with other people, and their day-to-day tasks looking after dependent family members living in the community. In this qualitative study, we explored their experiences and the coping strategies they applied, marked by a specific time axis, with vaccination being the turning point that differentiated “care during lockdown” from the “return to normality”. 

The research team identified five sub-themes within “care during lockdown”—the re-structuring of before-care services, the introduction of new care approaches, managing the dependent person’s health problems, looking after oneself, and dealing with adversity. They also looked at three sub-themes within “caregiving toward normality”—recovering confidence, recovering assistance, and reconnecting with others. 

A number of studies have identified similar themes [28]. Others [29] focused on the loss of control, uncertainty, and fear, which, throughout the interviews, our research team identified as crosscutting elements which have not defined their own categories, but which have influenced all of them. 

All the literature coincides in affirming an increase in the caregiving workload during the pandemic [28,30]. As far as the re-structuring of before-care services is concerned, in our research, we saw that many caregivers gave up their external assistance despite their increased caregiving workload. This topic has been studied in the United Kingdom [31] with contradictory results. Some caregivers cancelled their assistance, while others, despite their fear of contagion, could not do so, as they were unable to cover the care needs of the dependent person on their own. 

With respect to the work–life balance, our research highlighted two differing realities, which have also been described in another study within our field. Research has shown that working from home offers greater flexibility, alleviating some of the economic and financial difficulties that have arisen through the pandemic [30]. Others, such as the research undertaken by Borges-Machado in Portugal, describe the difficulty in carrying out normal work activities from the home in which the dependent person is being cared for, without having any time for relaxation and leisure [32]. 

The systematic review undertaken by Bailey et al. [30] highlights the fact that the main fear of caregivers was that the dependent person would become infected with COVID, prioritising that person’s safety above their own with the worries regarding who would look after them if they got sick. This fear also manifested itself in the interviews we analyzed in our research. Another point in common is that caregivers prolonged prevention measures beyond those applied by the general population. This included the idea that when other people broke the rules, this led to feelings of stress and frustration among caregivers. This feeling of uncertainty and insecurity that some of our caregivers manifested on assuming new care roles and responsibilities have also been identified in this and other studies [30,31]. 

Research undertaken in Europe highlights the negative impact on physical and mental health caused by the pandemic in those caring for family members [33]. Our caregivers’ discourses point to changes in their eating, exercise, and relaxation habits, with increased physical inactivity and time spent looking at the screens of computers and mobile devices, described by Greaney as an unequal impact between subjects [34]. In Spain, a population survey [20] highlighted the fact that caregivers showed higher levels of post-traumatic stress compared to non-caregivers. Despite the fact that this impact was not classified in the central axes of our research, symptoms of anxiety and stress were nevertheless identified. 

The use of electronic devices marked the daily routine of the population during the pandemic, as it provided the opportunity to maintain social contact without increasing the risk of contagion. In our research, we saw how access to this tool differed depending on the characteristics of the caregiver, with it being harder for older people with a lower socio-economic level. Ruksakulpiwat’s systematic review [33] also stressed the effects of the digital divide and the inequality that exists in terms of access to computer resources, which has meant that not everybody can benefit from these tools. This same research suggests that caregivers who had previously taken on caring on their own did not experience the effects of isolation as much, as they were used to keeping in touch with others via social media [33]. 

In line with other qualitative research, during the COVID-19 pandemic, caregivers said that while they had had less home care assistance and saw an increase in feelings of isolation, they also mentioned that family communication had improved [35]. Other research has noted that the relationship between the caregiver and the dependent person came under threat. Although we have not explored this aspect in any depth, we did not come across evidence of this in our interviews.

### 4.2. Strengths and Limitations 

This research covers a period of a year in the pandemic as experienced by those caring for family members in the community. It is the only study among the bibliography we consulted that places an analytical focus on the time axis of how reality and caregiving were transformed through the de-escalation of the pandemic. 

Most research carried out on caregiving in the time of COVID-19 focuses on the caregivers of people with dementia. However, those taking part in this study represented diversity in terms of their socio-demographic characteristics (gender, degree of relationship, age, and employment situation), as well as variety in the pathology of the dependents (dementia, physical disability, sensory disability, multimorbidity, etc.). 

Among the possible limitations, it should be remembered that we did not look at the families’ socioeconomic level despite employing purposive sampling. We were unable to include caregivers in rural areas, nor those who fell ill during lockdown or who had to stop providing care at any point. We did not find any material in this regard in the bibliography we consulted.

Another limitation we found in this study is that the results derived from this study could not be generalized to the caregivers of people who had active chemotherapy or radiotherapy treatment, as they were excluded from the sample. This decision was made considering that their experiences were likely very different from those of caregivers of dependent people.

### 4.3. Implications for Nursing Practice 

The self-reported experiences of caregivers in the pandemic context, the coping strategies identified, and the understanding of their vital experience highlight their resilience and adaptive abilities. These are elements to be taken into account by nurses in order to maintain the care of the individual at home and to provide the caregiver with a central role in self-management of their self-care. 

### 4.4. Health Policy Implications 

This study has highlighted the widely studied conditions of vulnerability of this group. It has shown the need for a health and social support system that must be able to learn from experience and adopt policies to be more accessible, more responsive, and more adaptable in its actions.

## 5. Conclusions

During the COVID-19 pandemic, there was an intensification of care for dependent individuals, particularly among those caring for family members without assistance. This period was marked by heightened fear, anxiety, feelings of loneliness, and uncertainty among caregivers. The focus shifted towards avoiding contagion and implementing self-care practices to continue providing care amidst adversity and after adapting to the ‘new normal’. Although vaccination initially caused mistrust among the general population, it seems to have been well-received by caregivers and dependent individuals, marking a turning point in the return to normalcy.

The study results highlight caregivers’ deep involvement in the care process, their emotional states, and their proactive approach to maintaining the health and safety of both themselves and their dependents. Considering the heterogeneity of contexts, experiences, and needs of caregivers, there is a greater emphasis on the need to individualize nursing care plans. This individualized attention should be accompanied by the design of personalized support programs and protocols for caregivers. These programs should encompass needs assessment, evaluation of the social context, available assistance, and resilience for the design of coordinated interventions within a multidisciplinary team. A multisectoral approach is necessary, where healthcare systems and social services converge to ensure that caregivers and dependent individuals do not face limited access to healthcare services in future pandemics.

Furthermore, this study highlights the crucial role of vaccination campaigns in protecting vulnerable populations. Vaccines played a crucial role in mitigating care risks during the pandemic, offering a positive outlook for future healthcare practices.

In conclusion, this study contributes to a deeper understanding of caregiving experiences during the pandemic and emphasizes the need for tailored support and policies to address the diverse needs of caregivers. It is imperative to continue exploring ways to empower and recognize the invaluable contributions of caregivers in healthcare settings.

## Figures and Tables

**Figure 1 healthcare-12-00970-f001:**
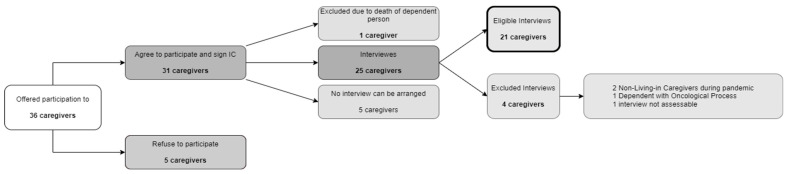
Flow diagram of participants.

**Table 1 healthcare-12-00970-t001:** Characteristics of family caregivers.

Interview	Age ^a^	Sex ^b^	Relationship	Employment Situation	Years Caring	Dependent Age ^a^	Dependent Disease
CJ_RSR_1	79	W	Wife	Retired	11	87	Alzheimer
CJ_RSR_6	28	M	Grandchild	Active	8	92	Immobilized
SAN_POR_1	66	W	Daughter	Retired	2	95	Ictus
SAN_POR_2	46	M	Son	Active	3	85	Poliosteoarthritis
SB_AMG_2	77	M	Husband	Retired	2	74	Alzheimer
SB_AMG_5	53	W	Daughter	Unemployment	4	92	Alzheimer
SB_AMG_8	46	W	Daughter	Active	2	78	Glaucoma
SB_AMG_9	56	W	Daughter	Domestic work	6	84	Alzheimer
SB_AMG_10	75	M	Husband	Retired	6	84	Alzheimer
BP_DGB_1	55	W	Daughter	Active		90	Parkinson
BP_DGB_2	55	W	Daughter	Active	14	86	Parkinson
BP_DGB_4	63	W	Daughter	Active	5	96	Alzheimer
CJ_RSR_12	67	W	Daughter-in-law	Retired	2	100	Cognitive impairment
CJ_RSR_13	47	W	Daughter	Unemployment	5	86	Glaucoma
CJ_RSR_7	57	W	Daughter	Active	5	86	Chronic renal disease
CJ_RSR_8	69	W	Daughter	Retired	6	91	Ictus
CJ_RSR_9	56	W	Daughter	Unemployment	8	96	Immobilized
CJ_SRS_11	67	W	Daughter	Retired	10	85	Cognitive impairment
SB_AMG_12	66	W	Daughter	Domestic work	2	95	Ictus
SAN_POR_7	60	W	Daughter	Active	6	92	Alzheimer
SB_AMG_1	77	W	Wife	Retired	11	78	Chronic renal disease

^a^ Age: years. ^b^ Sex: W = Woman; M = Male.

**Table 2 healthcare-12-00970-t002:** Themes and sub-themes identified on caregivers’ experiences during the pandemic.

Themes	Subthemes
Caregivers during lockdown—providing care in a time of adversity	Re-structuring family caregiving
	Assuming and incorporating new care
	Managing health problems
	Staying healthy and looking after oneself
	Dealing with changes
Caring towards normality	Recovering confidence
	The return of assistance
	Reconnecting with others

## Data Availability

The Ethics Committee approved this research without considering the option of data sharing or publication of interviews and recordings. The interviews contain sensitive information about the caregiver’s experiences, so there are ethical and legal restrictions to sharing the dataset. The interviews can be requested by contacting the main researcher. However, each new project based on these data must be previously submitted to Ethics Committee for approval.

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
