# Peer review of "Family Caregivers’ Experiences during the COVID-19 Pandemic: Qualitative Study"

_healthcare, 2024, doi:10.3390/healthcare12100970_

Round 1

Reviewer 1 Report

Comments and Suggestions for Authors

The article is well-written, theoretically well-founded and has a good discussion of the results. 

A few points for improvement:

In the "participants and sample" section, it should be made clearer how the nurse at the health centre selected the caregivers to be interviewed. It is said says that  ensure diversity should be ensure, but the selection process and who made the request/invitation for the interview should be better explained.

The limitations of the study should include the fact that carers of cancer patients were excluded, which should be justified.

The conclusion should be more detailed, in particular referring to the most significant topics of discussion and the implications for health policies. The statement that knowledge of carers' experiences and their adaptation promotes the participation of carers in the nursing process, made in the conclusion, needs to be explained and justified. 

Reviewer 2 Report

Comments and Suggestions for Authors

Thank you for the opportunity to review this paper. The introduction sets the scene nicely, giving sufficient contextual background.

Design – would expect to see reference to phenomenology, its import in this study and the most recent works of Braun and Clarke cited. Grounded Theory principles are mentioned in the participants and sampling, not referenced and relevance is unclear. Considering theoretical saturation, can the authors clarify how this was managed ethically? Did some participants who consented to participate not actually do so?

Results: Overall, the results are clear, there are some occasions where interpretation is not necessarily aligned with the supporting quotation. For example, I cannot see how “We would speak to each other [family members] on video calls” reflects the preservation of identity, this needs some consideration and development. The theme relating to confronting adversity might benefit from review, examples and quotations used seem at times to be more reflective of the staying healthy and looking after oneself – seeking solace, prayer, exercise, cognitive coping mechanisms (the quotation mentions “adapting the mind”, the reader might want to know how this is done). I wonder if the “dealing with adversity” term utilised in the paper would be better aligned with the findings?

Kindly review “Family caregivers who had keep working outside home altering their day-to-day  habits”, this is fragmented and unclear.

 The conclusions drawn do compare with other studies internationally. Whilst this is acceptable, the novelty of this study could be amplified by extrapolating the particular learnings related to the availability of the vaccinations  for example and offering some concrete thinking for future practice. For example, I would also like to see some development of the implications for nursing practice and health policy. How might nursing practice and health policy be more adaptive and responsive based on the findings presented?

Round 2

Reviewer 2 Report

Comments and Suggestions for Authors

The revisions made to the paper have strengthened the significance whereby readers can use findings to inform practice and deepen their understanding of the role of informal carers, their self-care/coping strategies beyond the pandemic and their contribution within formal healthcare settings.

Thank you for reconsidering the methods section, this is more transparent.